# The importance of radiation for semi-empirical water-use efficiency models

Sven Boese[1], Martin Jung[1], Nuno Carvalhais[1], Markus Reichstein[1]

[1]Max Planck Institute for Biogeochemistry, Dept. for Biogeochemical Integration, Hans-Knoell-Strasse 10, 07745 Jena

*Correspondence to*: Sven Boese (*sboese@bgc-jena.mpg.de*)

**Abstract**

Water-use efficiency (WUE) is a fundamental property for the coupling of carbon and water cycles in plants and ecosystems. Existing model formulations predicting this variable differ in the type of response of WUE to the atmospheric vapor pressure deficit of water (VPD). We tested a representative WUE model on ecosystem scale at 110 eddy-covariance sites of the

FLUXNET initiative by predicting evapotranspiration (ET) based on gross primary productivity (GPP) and VPD. We found that introducing an intercept term in the formulation increases model performance considerably, indicating that an additional factor needs to be considered. We demonstrate that this intercept term varies seasonally and we subsequently associate it with radiation. Replacing the constant intercept term with a linear function of global radiation was found to further improve model predictions of ET. Our new semi-empirical ecosystem WUE formulation indicates that, averaged over all sites, this

radiation term accounts for up to half (39–47 %) of transpiration. These empirical findings challenge the current understanding of water-use efficiency on ecosystem-scale.

## 1    Introduction

Understanding the coupling of carbon and water cycles is as a central question of global change research, as changes in one

of the cycles could directly propagate to the other (Churkina et al., 1999; Gerten et al., 2004; Ito & Inatomi, 2012). Carbon assimilation through photosynthesis ($A$) and transpiration ($T$) constitute major fluxes in these two cycles and the Earth system (Jasechko et al., 2013; Ciais et al., 2014). On the leaf-scale, *water-use efficiency* (WUE) is defined as WUE $= A/T$, quantifying the ability of a plant to assimilate atmospheric carbon dioxide per water loss by transpiration. An understanding of this ratio can hence translate into the ability to predict one of the two fluxes from the other.


Both fluxes are limited by the stomatal conductance ($g_s$) of plant leaves (Cowan and Farquhar, 1977), allowing for the definition of *intrinsic water-use efficiency*, WUE$_i = A/g_s$. This quantity is less contingent on the vapor pressure deficit of atmospheric water vapor (VPD) which affects both fluxes differently. At the ecosystem scale, neither carbon assimilation nor transpiration can be observed directly. However, the eddy-covariance (EC) method (Baldocchi et al., 2001) can be used to

obtain data of the gross-primary productivity (GPP) and the evapotranspiration (ET). On ecosystem scale, water-use efficiency is then defined as WUE = GPP/ET, while the intrinsic water-use-efficiency is accordingly formulated as $WUE_i =$ $GPP/G_s$, where $G_s$ denotes the surface conductance of the ecosystem.

Analytical models that predict WUE on the leaf-scale (Katul et al., 2010; Medlyn et al., 2011) were derived from theoretical optimality considerations. Corresponding variants were evaluated with ecosystem-scale flux measurements gathered by the FLUXNET[1] in empirical studies (Beer et al., 2009; Zhou et al., 2014; Zhou et al., 2015). The central difference between the existing models is the response of $g_s$ or $G_s$ to VPD, resulting in different dependencies of WUE on VPD. The concept of *inherent water-use efficiency* (IWUE) by Beer et al. (2009) corrects for the increasing diffusion of water vapor with higher

values of VPD. However, the proposed IWUE is still dependent on the difference between leaf-external and leaf-internal $CO_2$ concentrations and therefore stomatal conductance. Physiological models (e.g. Katul et al., 2010), suggest stomatal contraction with increasing VPD, as plants aim to minimize water loss. This was found to be an important factor on ecosystem scale as shown by Zhou et al. (2014) for half-hourly and by Zhou et al. (2015) for daily observations. In both studies, the response of WUE could be approximated to be proportional to $VPD^{-0.5}$. Despite their discrepancy, both the

models of Beer et al. and Zhou et al. imply that, at ecosystem-scale, WUE is strictly an inverse function of VPD or $VPD^{0.5}$, respectively.

Stomatal conductance models that were derived on plant scale include an additional conductance term $g_0$ that is invariable to changing levels of photosynthesis (Ball et al., 1987; Medlyn et al., 2011). Any transpiration resulting from this part of

stomatal conductance should be expected to be proportional to the gradient of the partial pressure of water vapor, quantified by the atmospheric VPD observations. In contrast to formulations commonly used in stand, land surface and vegetation models (de Kauwe et al., 2013), this process is not considered in current ecosystem-scale WUE formulations (Zhou et al. 2014, Zhou et al., 2015).

Incoming solar radiation is a driving factor for both photosynthesis and transpiration. However, existing WUE models posit that the ratio of both is invariant with regard to this factor. This implicitly assumes that, at ecosystem-scale, the influence of radiation on GPP and ET cancels out, as the water-use efficiency is entirely determined by gas diffusion and its limiting factors. However, photosynthesis saturates at high radiation levels (Farquhar et al., 1980), even in well-watered conditions. Models of Potential Evapotranspiration (PET), by contrast, do not prescribe a similar limiting behavior in conditions of

sufficient water-availability. For example, in the Penman–Monteith equation, the evaporation rate scales linearly with the absorbed radiation, holding all other factors constant (Leuning et al., 2008). Mechanistically, the process of equilibrium transpiration (Jarvis and McNaughton, 1986) implies that sizeable transpiration can occur even when the leaf is fully

---

[1] http://fluxnet.fluxdata.org/

decoupled from the atmosphere, i.e. when VPD is very low. This is a second process that current ecosystem WUE models cannot accommodate.

In this study, we address these unresolved inconsistencies regarding the importance of additional model terms for predictions of ecosystem-scale transpiration. We do this empirically on ecosystem-scale by optimizing and assessing different WUE models with FLUXNET observations from 110 globally distributed towers. In our approach, ET is selected as target variable, while the different WUE models utilize GPP as one of multiple explanatory variables. The substantial degree of correlation between GPP and ET is thus harnessed for the predictions of ET. In a first step, we identify existing biases in ecosystem-scale WUE models. In the next step, these biases are tested for their dependency on VPD and radiation. Lastly, we infer a tentative partitioning of transpiration according to its association with radiation and discuss the substantial magnitude of this metric. We conclude by highlighting how changes in the model structures impact the between-site variability of parameter estimates.

## 2    Data & methods

### 2.1    Data

The daily day-time integrals of gross primary productivity (GPP) and evapotranspiration (ET) were taken from the La Thuile FLUXNET (open and fair use data policy sites) collection[2]. The aggregation to day-time values was based on values of potential radiation larger than 10 W m$^{-2}$. Additionally, we used global radiation (Rg) and the day-time vapor pressure deficit (VPD) measured at the same eddy-covariance (EC) sites.

The EC data were processed according to the standard methods (Papale et al., 2006; Reichstein et al., 2005) to assure consistent quality of the observations. Eddy-covariance GPP results were based on the flux partitioning method of Reichstein (2005). We used only data with GPP > 0.1 gC d$^{-1}$ m$^{-2}$, ET > 0.05 mm d$^{-1}$ and VPD > 0.001 kPa) to reduce the relatively large impact of random measurement errors under low flux conditions.  Following the procedure of Beer et al. (2009) we further used only data after three consecutive rain-free days. This reduces contributions by evaporation to the measured evapotranspiration as physical evaporation typically declines rapidly after rain events due to the depletion of water stored in the topmost soil layer (Wythers et al., 1999). That assumption similarly applies to precipitation that is intercepted on leaf-surfaces and other plant parts in the canopy. As Miralles et al. (2010) summarize, the interception storage for forest ecosystems reported in different studies amounted to a mean of 1.2 mm (± 0.4 mm; per unit area of canopy cover). With the mean interception evaporation rates reported as 0.3 mm h$^{-1}$ (± 0.1 mm h$^{-1}$; per unit area of canopy cover), this storage can be concluded to be typically depleted within the first days after a precipitation event. Therefore, the measured

---

[2] FLUXNET Synthesis Dataset (La Thuile 2007). Available at www.fluxdata.org

evapotranspiration after three consecutive rain-free days is expected to approximate transpiration, and we additionally verified that our results are robust when considering longer rain-free periods (see below). For the main analysis, we included sites with at least 25 data-points fulfilling the requirements noted above. A list of these 110 sites used for the parameter estimation can be found in the supplement (Section S4, Table S2).

The presented analyses presume that the observed evapotranspiration is dominated by transpiration after three consecutive rain-free days. To test the robustness of our findings against this assumption, we varied the number of consecutive rain-free days from 0 to 14. For each considered step, the data were filtered accordingly and parameters of the WUE models were estimated for each site. We then calculated the mean of the parameter estimates over all sites and the associated uncertainty

10 (95% confidence interval) of the mean via bootstrapping. We excluded some humid sites from this sensitivity analysis for which too few data points were available when filtering for longer rain-free periods. This procedure ensures that all levels of filtering for rain-free periods included the same set of sites.

## 2.2 Concepts & models

For our analysis, we started out with the WUE model of Zhou et al. (2015), which we converted for an inversion against ET data:

$$ \mathrm{ET} \ = \ \frac{\mathrm{GPP} \cdot \mathrm{VPD}^{0.5}}{\mathrm{uWUE}} \ , \tag{1}$$

where uWUE represents the site-specific *underlying water-use efficiency*. The introduced WUE model states that no transpiration occurs independently from $\mathrm{GPP} \cdot \mathrm{VPD}^{0.5}$. To test this hypothesis, we introduced an intercept term ($\mathrm{ET_{res}}$) in the generic WUE model, which we call $+\mathrm{ET_{res}}$ model:

$$ \mathrm{ET} \ = \ \frac{\mathrm{GPP} \cdot \mathrm{VPD}^{0.5}}{\mathrm{uWUE}} + \mathrm{ET_{res}} \ . \tag{2}$$

Hence, any significant intercept would indicate transpiration that cannot be explained by $\mathrm{GPP} \cdot \mathrm{VPD}^{0.5}$. The relative contribution of $\mathrm{ET_{res}}$ to the mean predicted flux, $C_{\mathrm{ETres}}$, was calculated as:

$$ C_{\mathrm{ETres}} \ = \ \frac{\mathrm{ET_{res}}}{\frac{\mathrm{GPP} \cdot \mathrm{VPD}^{0.5}}{\mathrm{uWUE}} + \mathrm{ET_{res}}} \ , \tag{3}$$

where the denominator contains the mean predicted daily ET of the model.

To further test whether $ET_{res}$ relates to atmospheric variables, we postulated three different alternative models. The residual transpiration could be driven by an additional VPD term that is independent from photosynthesis, as some stomatal conductance models (Medlyn et al., 2011) already include a residual conductance term $g_0$, in which case the WUE model could take the form:

$$ET = \frac{GPP \cdot VPD^{0.5}}{uWUE} + g_0 \cdot VPD . \tag{4}$$

We also considered the possibility that $ET_{res}$ is related to global radiation (Rg):

$$ET = \frac{GPP \cdot VPD^{0.5}}{uWUE} + r \cdot Rg. \tag{5}$$

To test for possible interactions between the two additional variables, we considered a third alternative where $ET_{res}$ was modelled by included both Rg and VPD as independent factors:

$$ET = \frac{GPP \cdot VPD^{0.5}}{uWUE} + g_0 \cdot VPD + r \cdot Rg. \tag{6}$$

In the following, we refer to the reference WUE definition (Eq. 1) as "Zhou". We abbreviate models with additional covariates by omitting the reference to the GPP-VPD-term ($\frac{GPP \cdot VPD^{0.5}}{uWUE}$) of Zhou, which is always used unless denoted otherwise. The model with an additional VPD term, for example, is thus designated "+VPD".

## 2.3 Parameter estimation & statistics

In the following, we refer to Eq. 1, 2, 4, 5, 6 as models, as we optimize their fit to the EC data by estimating free parameters. The estimation was conducted with the Levenberg-Marquardt technique, minimizing the sum of squares of the model residuals. The residuals were calculated as the difference between observed and predicted ET. We used the nlsLM package in R (Elzhov *et al.*, 2015). All parameters were restricted to positive values, preventing biologically implausible estimates. The uncertainties and correlations of the parameters were calculated with the variance-covariance matrix returned by the fitting function (Omlin & Reichert, 1999).

The model performance was assessed for each site with the Nash-Sutcliffe model efficiency (MEF):

$$\text{MEF} = 1 - \frac{\sum(Y_{\text{prd}} - Y_{\text{obs}})^2}{\sum(Y_{\text{obs}} - \overline{Y_{\text{obs}}})^2},$$

(7)

where $Y_{\text{obs}}$ are observations and $Y_{\text{prd}}$ are predictions by a model.

A MEF of 1 implies a perfect fit of the model to the data. A MEF below 0 implies that the mean of the observations outperforms the fit of the model. All MEFs were calculated in a *leave-one-out* cross-validation to account for the problem of
over-fitting. Thus, the cross-validated MEFs can be used to compare models with differing numbers of free parameters.

We assessed parameter distributions stratified according to a classification of vegetation structure that was based on plant functional types (PFT). The included FLUXNET sites were classified as *low vegetation structure* if the report PFT was either grassland (GRA) or crops (CRO) and *high vegetation structure* if otherwise. In total, 40 sites had *low vegetation*
*structure*, compared to 70 sites with *high vegetation structure*. The significance of differences between distributions of parameters and metrics for these two classes was verified by a Kolmogorov–Smirnov test (Daniel, 1990) and bootstrapped 95% confidence intervals of the mean.

## 2.4   Contribution Analysis

To assess the contribution of driving variables to the predicted fluxes, we performed an analysis of attribution to the individual model terms. Consider a simple multiple linear regression model,

$$Y = a_1 \cdot X_1 + a_2 \cdot X_2,$$

(8)


where $Y$ is the dependent variable, $X_1$ and $X_2$ are independent variables and $a_1$ and $a_2$ denote the model parameters. Due to the additive character of the model, the contribution of one variable (e.g. $X_1$) to the total flux is given by its product with the slope ($a \cdot X_1$). In our analysis, we tested the contribution of the linear radiation term (Eq. 5) to the total modelled evapotranspiration. Thus we defined the fraction of evapotranspiration that was attributed to the radiation term as:


$$\text{ET}_{\text{frac}} = \frac{\sum r \cdot \text{Rg}}{\sum\left(\frac{\text{GPP} \cdot \sqrt{\text{VPD}}}{\text{uWUE}} + r \cdot \text{Rg}\right)}$$

(9)

We considered two variants to estimate this metric: *parallel* and *hierarchical*. In the first case, both parameters, uWUE and $r$ were estimated in a standard parameter estimation, i.e. concurrently. In the second case, uWUE was first estimated in the $+\text{ET}_{\text{res}}$ model. We then defined a term $M$ as:

$$M = \text{ET}_{\text{res}} + \varepsilon, \tag{10}$$

where $\varepsilon$ denotes the residuals and $\text{ET}_{\text{res}}$ denotes the intercept parameter of the $+\text{ET}_{\text{res}}$ model. The parameter $r$ was then estimated in a linear regression of the form:

$$M = r \cdot Rg + c, \tag{11}$$

where $c$ denotes a constant intercept. By giving precedence to the uWUE parameter in this approach, we expect to get an estimate of a reasonable lower bound for $\text{ET}_{\text{frac}}$. All parameters were constrained to positive values.

To further assess uncertainties due to problems of parameter identifiability among uWUE and $r$ in the *parallel* variant, we sampled 200 parameter vectors from the posterior parameter uncertainty distribution for each site.

*Impact of parameter correlations on the contribution analysis*

In many realistic examples, the model parameters of Eq. 8, $a_1$ and $a_2$, are not perfectly identifiable. This could be due to the correlation of $X_1$ and $X_2$ or more fundamental model structural uncertainty. In these cases, $f_{X_i}$ is confounded by the parameter correlation $a_1$ and $a_2$, leading to a high uncertainty of its estimation. In our analysis, GPP and Rg are expected to be highly correlated, leading to dependent parameter uncertainties. To evaluate this effect, we estimated the contribution of the described within-site uncertainty of $f_{X_i}$. The variance-covariance matrix ($V$) of the parameter estimates could be calculated for each site with the results of the regression. Consequently, $V$ can be used to derive the respective posterior parameter distributions, from which we sampled 200 parameter vectors per site, representing the uncertainty and correlation of the two parameters. Site-specific vectors $f_{X_i,S}$ can be calculated as

$$f_{X_i,s_j} = \left[ f_{X_i,s_j,p_1}, f_{X_i,s_j,p_2}, \dots, f_{X_i,s_j,p_{200}} \right], \tag{12}$$

where $p_1$ to $p_{200}$ denote the 200 realizations of parameter vectors and $s_j$ denotes a specific site.

Based on this, we tested whether the global variance of $f_{X_i}$ is a product of differing means of the site-specific $f_{X_i,S}$ vectors or their variances. For this, we performed an ANOVA along the sites as categorical variable with the parameter realizations as random replicates.

## 3    Results

In our analysis we tested different water-use efficiency (WUE) models that predicted evapotranspiration (ET) using the product of gross-primary productivity (GPP) and the water vapor pressure deficit, $GPP \cdot VPD^{0.5}$, as predictor variable. When plotting ET as a function of this multiplicative term, we observed significant intercepts, e.g. for the Mediterranean FLUXNET site IT-BCi (Fig. 1a). In these cases, significant ET was observed when the driving force of the established models, the GPP-VPD-product, was small or zero. When we explicitly included this term in the model (+ETres), the cross-validated MEF increased notably (Fig. 1b). As Table 1 shows, the +ETres variant outperformed the Zhou model at 86% of the sites. The respective mean difference in MEF between the two variants was 0.07 (Table 2). A corresponding table that presents the mean difference of the root mean squared error (RMSE) for all sites can be found in the supplementary materials (Table S1).

**Table 1.  Fraction of the 110 sites at which the model of the respective row was superior to the model of the respective column according to the MEF$_{diff}$ criterion.**

|         | Zhou | +ETres | +VPD | +Rg  | +VPD+Rg |
|---------|------|--------|------|------|---------|
| Zhou    | -    | 0.14   | 0.22 | 0.07 | 0.07    |
| +ETres  | 0.86 | -      | 0.43 | 0.07 | 0.08    |
| +VPD    | 0.78 | 0.57   | -    | 0.08 | 0.08    |
| +Rg     | 0.93 | 0.93   | 0.92 | -    | 0.71    |
| +VPD+Rg | 0.93 | 0.92   | 0.92 | 0.29 | -       |

25

**Table 2. Mean MEF$_{diff}$ comparison for different model variants. Entries indicate the mean of the difference MEF(*row model*) − MEF(*column model*) for all 110 sites.**

|          | Zhou  | +ETres | +VPD  | +Rg   | +VPD+Rg |
|----------|-------|--------|-------|-------|---------|
| Zhou     | -     | -0.07  | -0.09 | -0.16 | -0.17   |
| +ETres   | 0.07  | -      | -0.02 | -0.09 | -0.10   |
| +VPD     | 0.09  | 0.02   | -     | -0.07 | -0.07   |
| +Rg      | 0.16  | 0.09   | 0.07  | -     | 0.00    |
| +VPD+Rg  | 0.17  | 0.10   | 0.07  | 0.00  | -       |

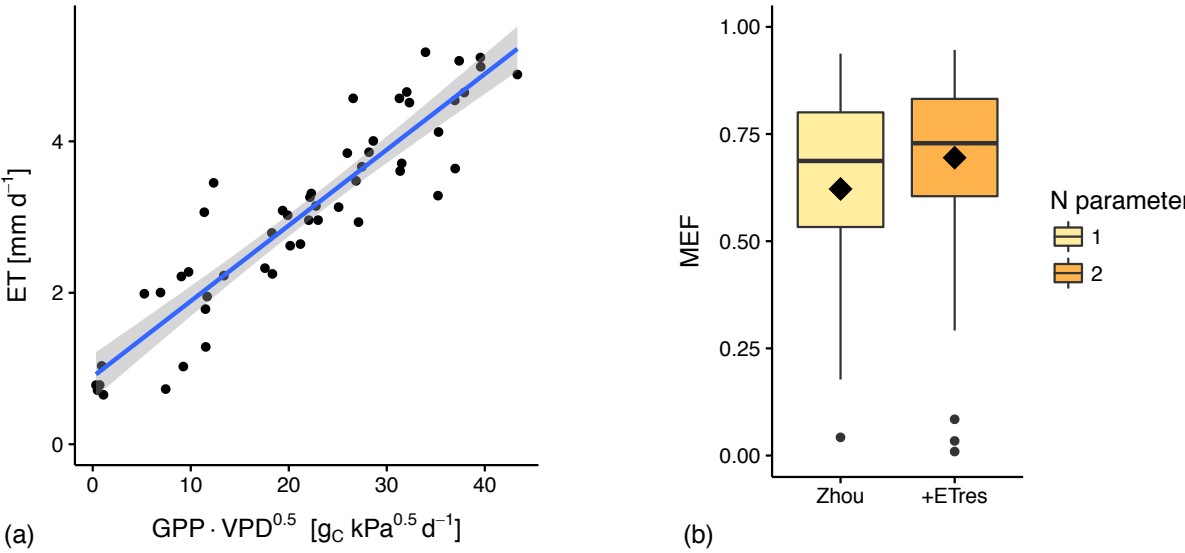

(a)  (b)

Figure 1. Linear regression between the product of GPP and VPD$^{0.5}$ and the daily ET; the shaded area indicates the 95% confidence interval of the mean of the predictions (a). Model performance of the Zhou model compared to the +ET$_{res}$ variant (b). The thick horizontal lines of the boxplots denote the median, the diamonds denote the mean of the MEF of all sites.

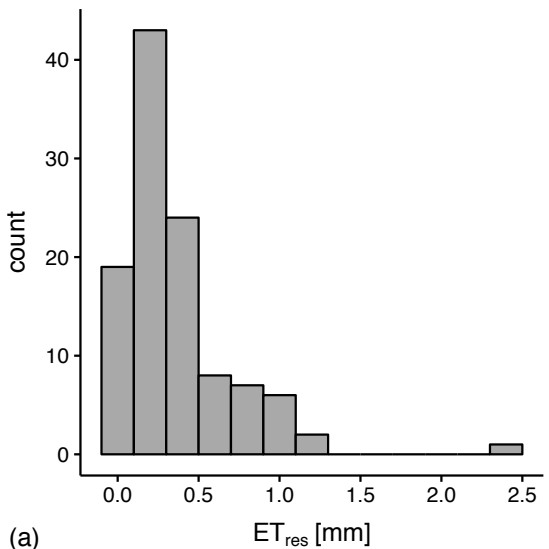 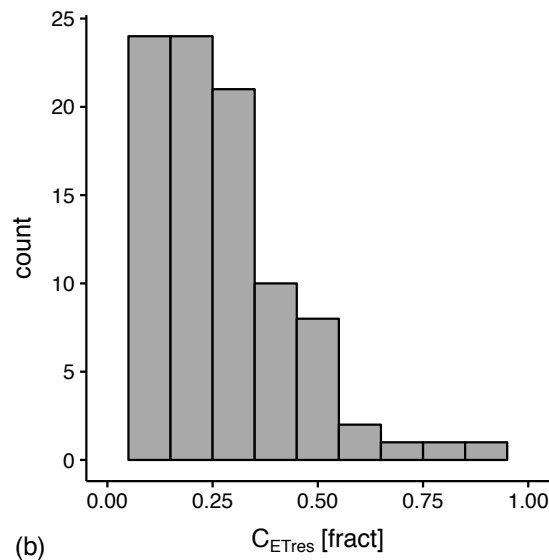

**Figure 2. Distribution of the estimated intercepts. The majority of sites had values between 0–1.2 mm for the absolute intercept (a). The relative contribution of ET$_{res}$ to the daily mean evapotranspiration as quantified by $C_{ETres}$ reached up to 0.94 (b).**

Of the 110 sites included in the analysis, 78% had a significant intercept. The site specifically estimated intercept values ranged from 0 to 2.36 mm (90%-percentile 0.86 mm) with a mean of 0.36 mm (Fig. 2a). The relative intercept $C_{ETres}$ reflects the relative magnitude of the intercept to the mean daily ET of the respective sites. It ranged from 0 to 0.86 (90%-percentile 0.46) with a mean of 0.23. This implies that circa a quarter of transpiration was not attributed to $GPP \cdot VPD^{0.5}$ in this model formulation (Fig. 2b). The importance of the intercept for the prediction of daily ET and its diverging values raise the

question whether the intercept compensates for the absence of a physical or biological process in the model or whether the observations are confounded by a systematic problem.

Our first hypothesis was that the ET$_{res}$ intercept was due to the remaining contributions of soil and interception evaporation to measured ET after three consecutive rain-free days. To test this, we estimated the parameter ET$_{res}$ for periods of

successively longer consecutive rain-free days. If our hypothesis was right, we would expect a trend of declining ET$_{res}$ with increasing consecutive rain-free days. However, no reduction of ET$_{res}$ beyond the exclusion of the three days after precipitation proposed by Beer et al. (dotted line) could be observed (Fig. 3a). We therefore concluded that potential contributions of soil and interception evaporation to ET cannot explain the existence of the intercept term in the WUE model.

We then hypothesized that a missing process in the model would be discernible as a temporal pattern in the $ET_{res}$ estimates. For that, we estimated the intercept for each month and site separately. The monthly means of the intercept for all sites varied in a clear seasonal pattern (Fig. 3b).[3]

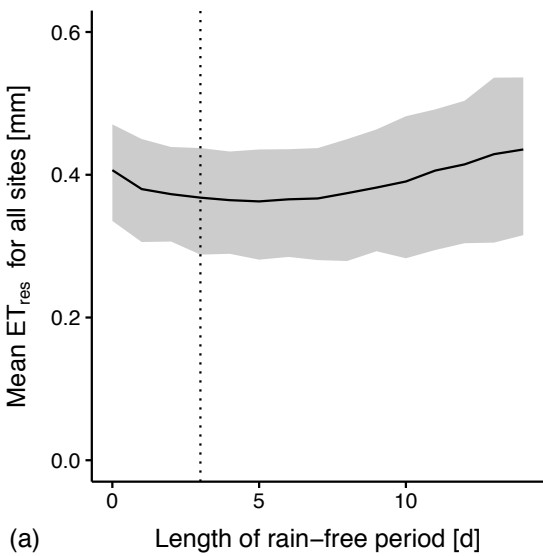 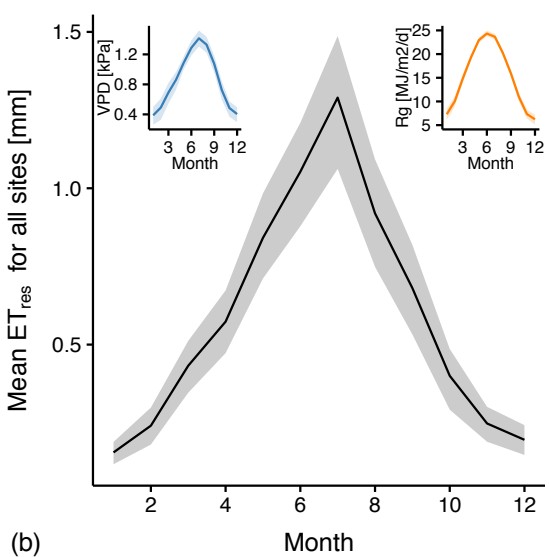

(a)  Length of rain−free period [d]    (b)  Month

**Figure 3. Sensitivity of $ET_{res}$ to varying length of filtering after precipitation events (a). The dotted, vertical line indicates the three-day period we adopted from Beer *et al*.  Monthly estimation of the intercept parameter $ET_{res}$. The inset plots illustrate the mean seasonal variability of VPD and Rg for all sites on a monthly scale (b). In all plots, the solid line is the mean for all sites; the band denotes the 95% confidence interval of the mean derived by bootstrapping.**

The seasonality of $ET_{res}$ suggests a relationship to meteorological variables such as VPD or radiation that vary seasonally too. A relationship with VPD could represent the $g_0$ term of canopy conductance models (Ball et al., 1987). It was therefore introduced in the form of an additional linear term (Eq. 4). A relationship with global radiation (Rg) (Eq. 5) could represent *equilibrium transpiration* (Jarvis and McNaughton, 1986), where the energy surplus of incoming radiation forces a transpirational flux independent from the vapor pressure gradient. We tested modeling the $ET_{res}$ intercept by including both variables separately (Eq. 4, 5) and jointly (Eq. 6).

We found only a small performance increase of the +VPD variant with regard to the Zhou model with intercept term ("+ETres", Fig. 4a). In fact, the $MEF_{diff}$ suggested that the +$ET_{res}$ model was still superior to the +VPD variant at 43% of the

---

[3] The monthly variability of $ET_{res}$ for all sites individually can be found in the supplementary materials (Section S2, Fig. S1).

sites (Table 1). By comparison, the +Rg model showed a substantial mean increase of 0.09 in the MEF compared to the +ET$_{res}$ model (Table 2). The increased model complexity (two free parameters) was justified at 93% of the sites. The model variant with both VPD and Rg terms ("+VPD+Rg") did not improve predictions compared to the simpler +Rg variant . Plots showing the impact of the +ET$_{res}$ and +Rg variant on the accuracy of the ET predictions for two selected sites can be found in the supplementary materials (Section S3, Fig. S2,S3)

We also tested whether the variation of ET that was unexplained by the +Rg model was still correlated with radiation. This would indicate that the chosen linear radiation-term did not fully account for all covariation between ET and Rg, implying a nonlinear dependency. The mean $R^2$ between the residuals of the +Rg variant and observed ET was 3.9% (maximum: 22%, 90%-percentile: 10%). This suggests that the linear variant was an adequate choice, as little of the unexplained variation of ET was still correlated with radiation.

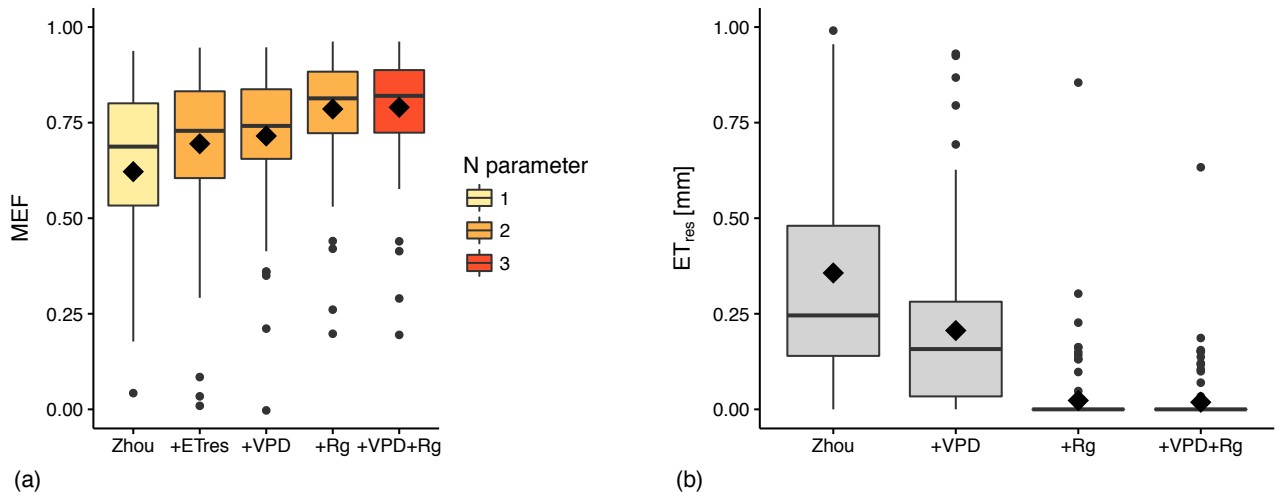

(a)                                                                                            (b)

**Figure 4. Cross-validated MEFs of the model variants with additional covariates (a). The thick horizontal lines of the boxplots denote the median, the diamonds denote the mean of the MEF of all sites. The +Rg variant lead to a further increase of model performance when comparing to the +ET$_{res}$ variant. By contrast, the +VPD variant showed only a small increase of performance. Global distribution of the residual intercept for all model variants (b).**

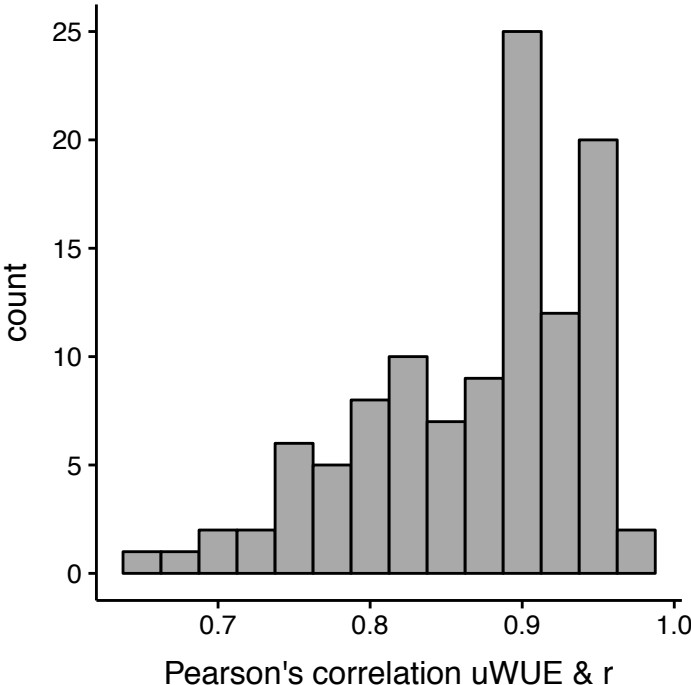

**Figure 5. Correlation of the parameter estimates for *r* and uWUE across sites.**

The difference between the +Rg and +VPD model is equally distinct, when the presence of a remaining model bias was tested. For this, the intercept was estimated with all four model variants (Zhou, +VPD, +Rg, +VPD+Rg). Only the models with an Rg-term had considerably reduced residual intercepts (Fig. 4b).

All models compared to the original definition Zhou had two or more parameters. Of those models with two parameters, the +Rg emerged as the best model after cross-validation, indicating that the additional model complexity was in fact justified. The presence of two parameters raised the question whether they could be identified independently. In fact, we found a high degree of correlation between the parameters for all sites (Fig. 5). It is likely that the correlation originates from the correlation between GPP and Rg.

Parameter distributions were separated for low and high vegetation structure (Fig. 6). The estimates of uWUE (a) were not significantly different for low and high vegetation (Kolmogorov–Smirnov test, p =0.35). By contrast, the estimates for r (b) were significantly higher for low (mean: 0.054, 95%-CI: 0.047–0.062) compared to high vegetation structure (mean: 0.041, 95%-CI: 0.034–0.049; Kolmogorov–Smirnov test, p = 0.003).

As transpiration of the +Rg model is a linear combination of stomatal and radiation-driven components, it is possible to calculate the relative contribution of each component to daily ET fluxes. We compared two approaches to calculate this quantitiy: *Parallel* for estimating both parameters concurrently, *hierarchical* for estimating $r$ only after uWUE has been calibrated (Eq. 10, 11). In both approaches, we observed that a sizeable fraction of mean daily ET could be attributed to the radiation term (Fig. 7a). For the *hierarchical* approach, the mean global $ET_{frac}$ was 24% (95%-CI: 21–27%); for the *parallel* approach, the mean global $ET_{frac}$ was 43% (95%-CI: 39–47%). Similar to the assessment of parameter distributions, we stratified the $ET_{frac}$ index for low and high vegetation structures (Fig. 7b). $ET_{frac}$ was significantly higher for low (mean: 0.53; 95%-CI: 0.48–0.58) compared to high vegetation structure (mean: 0.38; 95%-CI: 0.33–0.43; Kolmogorov–Smirnoff test p < 0.001).

Notably, $ET_{frac}$ estimated with the *parallel* approach varied widely between the sites (Fig. 8a). We assessed to which degree this variability can be interpreted as between-site variability of the expected value of $ET_{frac}$ or whether it is due to poorly constrained and correlated parameters due to collinearity (Fig. 5). The 'within-site' uncertainty of $ET_{frac}$ caused by parameter uncertainty was quantified as the range between the 97.5 and the 2.5 percentiles (95% confidence interval, CI) of $ET_{frac}$ estimates from 200 parameter vectors sampled from their respective posterior distributions. The vast majority of sites had an CI lower than 0.3 (Fig. 8b). This suggests that the large variability of $ET_{frac}$ was not a result of parameter uncertainties. The conducted ANOVA supported this conclusion, as it revealed that 96% of the global $ET_{frac}$ variability could be attributed to the variability between sites.

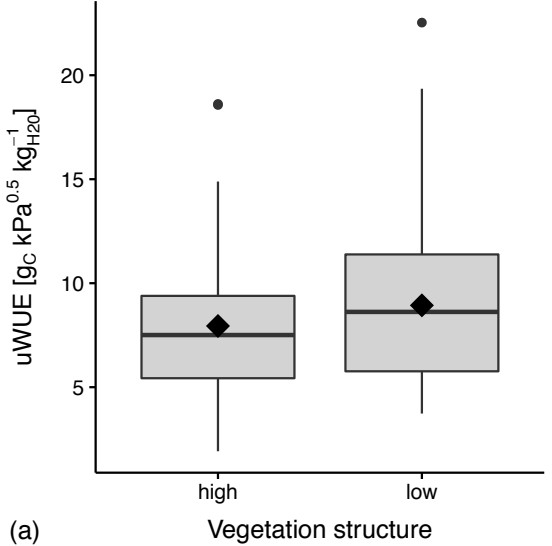

(a)

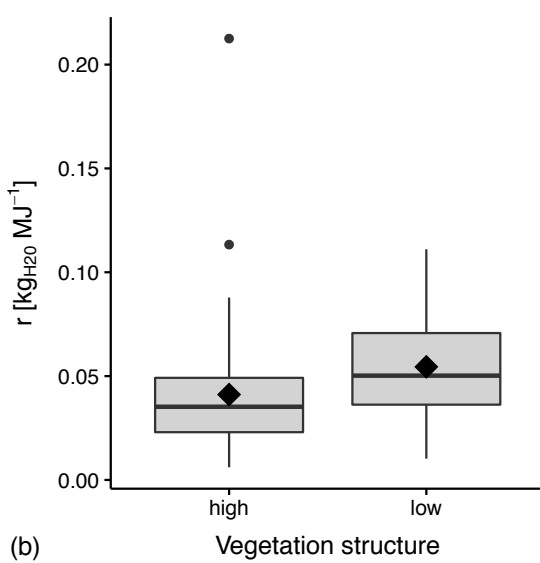

(b)

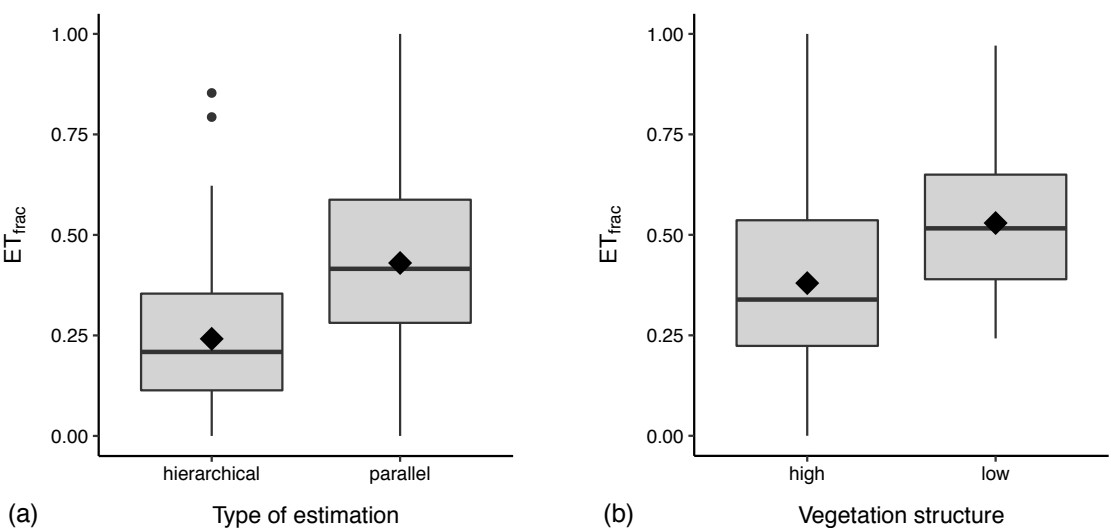

**Fig. 6. Distribution of the parameter estimates partitioned by type of vegetation structure. For uWUE (a), a Kolmogorov–Smirnov test yielded no difference in the distribution between the two classes of vegetation structure (p = 0.35). For *r* (b), the same test indicated a difference in the distribution between the two classes of vegetation structure (p = 0.003).**

**Fig. 7. Distributions of $ET_{frac}$ estimates for all sites. The diamonds indicate the mean, the bold horizontal lines indicate the median. The hierarchical approach yielded substantially lower values and can be interpreted as a conservative estimate of the quantity (a). $ET_{frac}$ was significantly higher in sites with low vegetation structure (b).**

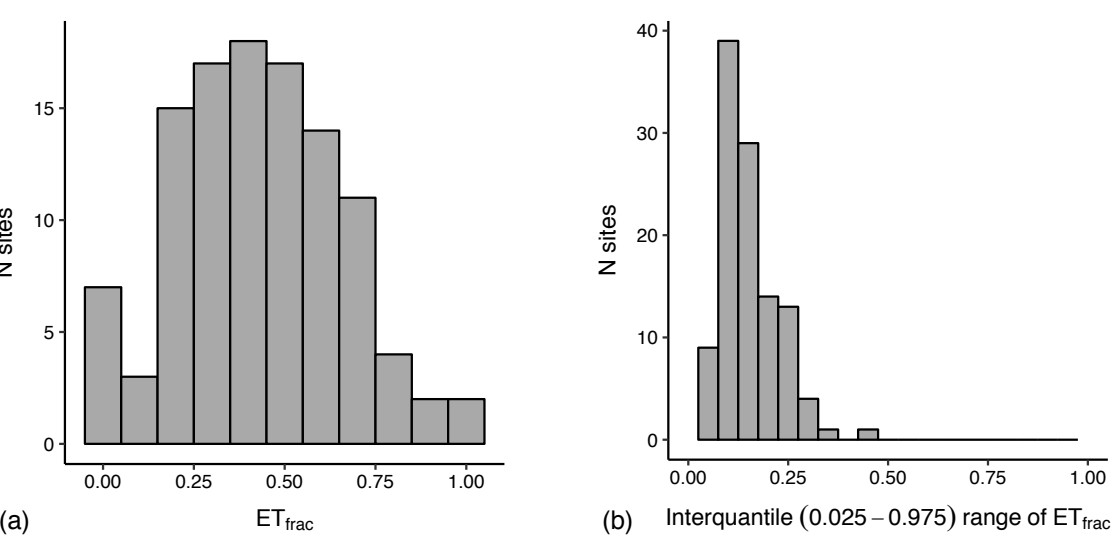

**Figure 8ab.** **Distribution of the fraction of mean daily transpiration (ET$_{frac}$) attributed to the radiation term for all sites, calculated with the *parallel* approach (a). Distribution of the within-site variability of ET$_{frac}$, calculated as the difference between the 97.5 and 2.5 percentiles of the ET$_{frac}$ estimates derived by sampling from the posterior parameter densities, parallel approach (b).**

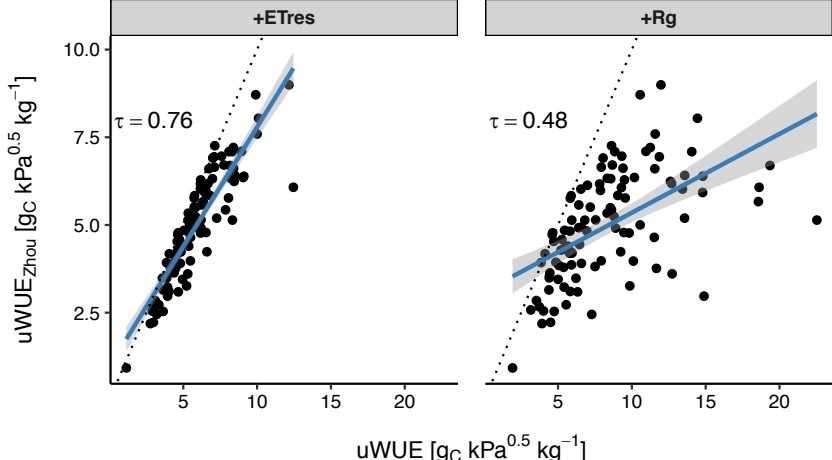

**Figure 9. Effect of different model variants on global uWUE estimates. The 1:1 line is dotted. A linear dependency would indicate that uWUE estimates are commensurable across sites, even if only after a linear scaling. When comparing with the estimates of the original Zhou model, the correlation is notably weaker for +Rg than for +ET$_{res}$. Sites with high parameter uncertainties were**
**removed for all models.**

Finally, we assessed the impact of ET$_{res}$ and Rg as additional covariates on the global variability of uWUE estimates (Fig. 9). We calculated Kendall's τ rank correlation coefficient (Kendall, 1938) between the site-level estimates of uWUEs derived from the Zhou model and two different variants: +ET$_{res}$ and +Rg. The degree of correlation of the uWUE estimates

quantifies whether changes in the model structure permutes the ordering of the estimates between sites. The extent to which parameter estimates are affected by the model structure is crucial because any explanation or prediction of parameter values between sites would be highly desirable. While a moderate correlation between the uWUE estimates of the Zhou model with the +ET$_{res}$ variant can be seen (τ = 0.76), the correlation of the uWUE estimates of the Zhou model with the respective values of the +Rg variant is low (τ = 0.48). The between-site variability of uWUE explained by the estimates of the Zhou

variant was 79% for the +ET$_{res}$ variant and 31% for the +Rg variant, as quantified by the $R^2$.

## 4    Discussion

### 4.1    Findings and mechanisms

In this study, we identified radiation as an important variable for ecosystem-scale transpiration and water-use efficiency.
Depending on the approach used, we attributed between a quarter and half of mean daily transpiration of all included
FLUXNET sites to a linear radiation term. These findings raise the question which biophysical or ecophysiological processes
can account for the estimated magnitudes of this attributed fraction.

The influence of radiation on stomatal conductance has been noted and discussed in the literature (Whitehead et al., 1981;
Jarvis and McNaughton, 1986) and is also reflected in existing transpiration models (Leuning et al., 2008). By contrast, we
detected a substantial transpiration component that was statistically independent from the product of GPP and $VPD^{0.5}$,
tentatively suggesting an insensitivity to stomatal conductance. Our results suggest that this additional flux could not be
associated with a $g_0$ conductance term, as this would imply the dependency of the additional ET on a linear VPD term (Ball
et al., 1987). By contrast, the intercept was shown to be more consistent with radiation. Consequently, models that integrated
radiation-driven transpiration with such an additive linear response had a superior predictive performance across flux towers.

The observed effect of radiation indicates that equilibrium transpiration could play an important role in ecosystem-scale
transpiration. Equilibrium transpiration (or equilibrium evaporation rate) is the transpiration occurring if the leaf is
completely decoupled from the atmosphere (Jarvis and McNaughton, 1986). Therefore, equilibrium transpiration is
independent from stomatal conductance and driven solely by the dissipation of energy, provided by incoming solar radiation.
According to McNaughton and Jarvis (1983), the decoupling parameter $\Omega$ is notably higher in grasslands compared to
forests. This parameter reflects the proportion of evaporation that is independent from the driving gradient in the water
vapour pressure. Our observation that both the radiation parameter $r$ and the index $ET_{frac}$ were significantly higher for
ecosystems with low vegetation structure is therefore consistent with the explanation that equilibrium evaporation is
responsible for the observed role of radiation.

This can be contrasted with a competing explanation that the radiation dependency of transpiration does not reflect an
additional process but rather a systematic problem of the VPD observations. VPD is measured together with the fluxes above
the canopy. While the recorded water and carbon fluxes do in fact represent the net fluxes of the tower footprint, the same
cannot be said for VPD. Its measurement above the canopy may differ substantially from the relevant magnitude of the
variable at the leaf-scale. However, leaf temperature and thus the VPD of the leaf boundary layer is dependent on solar
radiation (Tenhunen et al., 1990). Adding solar radiation to the equation could therefore be seen as compensating for the lack
of the aforementioned leaf-scale VPD observations.

## 4.2    Limitations

One limitation stems from the selection of rain-free periods for the parameter estimation. As previously described, this is a necessary step to justifiably assume that the observed latent heat fluxes constitute mostly a transpiration flux, rather than evaporation from bare-soil and leaf surfaces in addition to transpiration. It also makes our work comparable with the study of Beer et al. (2009) that used this method to derive their estimates. For the observed residual evapotranspiration $ET_{res}$ we could show that it is not an artefact of insufficient exclusion of days after precipitation events. However, the environmental conditions during and after rain events generally represent some of these specific conditions: Low VPD due to the moisture available for evaporation, a higher share of diffuse radiation and a more or less sudden increase in soil moisture among others. All mentioned variables could plausibly be assumed to have an influence on the stomatal opening of plants. Therefore, the presented WUE model must not necessarily predict flux relationships during or immediately after precipitation events due to the underrepresentation of similar conditions in the sample of observations used for parameter estimation.

The effect of atmospheric $CO_2$ concentrations on WUE (Morison, 1985; Conley et al., 2001) was not considered in this analysis. One could suspect that the seasonal variability of $ET_{res}$ is affected by seasonal $CO_2$ variability if both were in phase. However, $ET_{res}$ showed a global maximum during June–July, while northern hemisphere $CO_2$ concentrations are at its minimum in September–October (Keeling et al., 1976), implying that $CO_2$ concentrations are unlikely to cause the seasonal variation of $ET_{res}$.

The presented results are subject to the eco-climatological representativeness of the FLUXNET in general and sites with a *fair-and-free-use* data policy specifically. This means that tropical areas in particular are underrepresented in our analysis (with only one Australian site, AU-How). However, all posited mechanisms that could be responsible for the observed effect of radiation are not expected to be restricted to extra-tropical regions. Further research is however required to quantify the importance of radiation for water-use efficiency and transpiration in tropical ecosystems.

The model structures we tested in this study were evaluated according to their global empirical adequacy. Thus, despite the possible identification of probable mechanisms responsible for the observed patterns, the model structure selected in the end likely does not reflect the exact physical mechanisms by which the ecosystem operates. Furthermore, this limitation can be specifically important in the case of water limitation. As the Rg-term is completely independent from any variable reflecting vegetation activity, our model would predict transpiration scaling with radiation during periods of severe drought. By principle, the same problem could affect periods of low temperatures before leaf flushing, although these periods are generally also associated with low radiation levels and hence may not be as problematic.

## 4.3    Implications

Our empirical analysis suggests that ecosystem-scale transpiration depends to a sizeable degree on radiation rather than only the product GPP and $VPD^{0.5}$. This implies that photosynthesis and transpiration might be less strongly coupled on ecosystem scale than commonly assumed. We speculate that the additional effect of radiation could be due to equilibrium transpiration where radiation drives transpiration even when the canopy is fully decoupled from the atmosphere. The provided evidence for the differences between low and high vegetation structures gives additional credence to this explanation. However, we cannot disentangle direct physiological effects of radiation on transpiration from other radiation effects that are relevant on ecosystem scale at this point. For the latter, leaf to canopy scaling, micrometeorological conditions, and boundary layer dynamics might contribute to the observed relationship between ecosystem scale water use efficiency and radiation. Thus, further research is needed to reconcile our empirical findings with detailed ecosystem-scale modeling and theory on the one hand and plant-physiological research under controlled conditions on the other hand.

Finally, we caution against prematurely interpreting the between-site variability of underlying water-use efficiency (uWUE). We showed that estimates of uWUE derived with the Zhou model explained only a third of the observed variability of ecosystem uWUE derived from our empirically superior model formulation. The dependence of uWUE estimates on the chosen model formulation makes the interpretation of uWUE as an ecosystem property problematic. This concern would also hold for assessing temporal dynamics of uWUE such as long-term trends. For example, the unexpectedly large global trend of WUE across FLUXNET sites (Keenan et al., 2013) would need to be tested for its omission of radiation in the model that was used. Overall, this study highlights the importance of model structure uncertainty for interpretations of parameter variability.

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

**Data Availability**

For this study, we used observations of the FLUXNET initiative from sites with an *open and fair use data policy*[4]. The data sets can be downloaded at http://fluxnet.fluxdata.org//data/download-data/

5   **Acknowledgements**

We would like to thank the principal investigators of the FLUXNET initiative for providing the data used in this analysis. We are grateful to Anke Hildebrandt for discussion and feedback on the manuscript.

---

[4] http://www.fluxdata.org/Shared%20Documents/Policy_Free_Final.pdf