# Peer review of "The importance of radiation for semi-empirical water-use efficiency models"

_Biogeosciences, 2016_

## Referee Comment (RC1) · Anonymous Referee #1 · 25 Jan 2017

Boese et al. present empirical evidence of an intercept term in the formulation of the WUE model at ecosystem scale.

The WUE models at plant scale include a non-zero intercept term ment to represent the existence of conductance under very low VPD conditions or no photosynthesis. The canopy or ecosystem-level models however do not present such intercept. The first step was to demonstrate, based on EC measurements over X sites, that the introduction of an intercept would improve the ecosystem-level WUE model. The second step was to show the existence of a seasonality in the intercept, which proved to be related to the radiation.

The main results of the study are that radiation has an influential role on the WUE, and the formulation of a new ecosystem-level ET/WUE model. The manuscript is well

written and makes a clear and substantiated demonstration of the new dependencies evidenced, and their consequences.

Specific comments

The statistical method has been treated with great care. The use of cross-validated MEF to compare models with different forms and number of parameters was a good choice. But MEF does not provide a quantitative measure of the increase in the amount of variance explained by the successive modifications to the model. Table 2 could contain such an estimation, based i.e. on the (average) changes in the RMSE. The only problem in the model definition is that the intercept is left to be negative, which has no biological meaning and probably occurs because of a particular distribution of the observations in some sites. Following the idea that the intercept represents a low-VPD conductance, a negative but significant intercept should not be counted even if there is an overall model improvement. To circumvent this issue the hard way, the intercept parameter had to be represented as a squared term: $Y = a.X + Z^2 + epsilon$, where $Z^2 = ETres$. Changing this would suppose to redo many tables and figures. So for simplicity, the occurrences of negative intercepts can be subtracted from the tables 1, 2 and Fig 1 with a mention that models with negative intercepts were not used -as done for instance in Fig. 8.

It remains unclear why the effect of the radiation was modelled as a linear (P10 L7-10). A graph would help. Overall the manuscript has the tendency to not display the data and the relationships between them. Showing, for a couple of examples, the gain in having an intercept and incorporating radiation in the modelling would be great. This could be done in the form of time series. The number of figures included in the manuscript is already large and this figure could come in the supplementary material as a complement.

Minor comments

There are diverse typos and mistakes. Nothing major, the manuscript is well written

and easy to read.

P3L 15-20. How many times (or in percent) has the intercept ETres been found significant?

P3L13. Eddy-covariance is misspelled

P3L19 and hereafter. 'rain free' should be spelled uniformly throughout the manuscript, and I would use the hyphenation for ease of reading as in P4L2.

P4L6: unfinished sentence! " This procedure can thus ensure that. . ."

P6L9 The subsection's title "Partitioning of linear models" does not represent the content of the section which explains the methods used to estimate the contribution of each term in model. Thus Contribution partitioning or estimation of the contribution of models' components could be envisaged.

P6L 21 In the equation 9 it is not clear that the sum refers to the entire denominator, brackets should be used to avoid any misreading: $\sum ET + r.Rg$

would become $\sum (ET + r.Rg)$.

P7L 15. The variable STO has not been introduced!

P9L7. space needed between 0.34 and mm.

L8. Cut "is" in "a quarter of transpiration is was not. . .".

L11. Insert 'the' in ". . .was due to remaining contributions of. . .".

L19. cut 'representation' in "a missing process representation in the model. . .".

P10L7. replace 'parameter' by 'ETres' in "The seasonality of the parameter suggests. . .".

P10L9. Replace 'an' by the form of an in "It was therefore introduced in an additional. . ."

Table 2. Indicate the number of sites used to compute the mean MEF. Is the number

constant between variants?

---

## Referee Comment (RC2) · Anonymous Referee #2 · 4 Feb 2017

The article introduces a simple semi-empirical WUE model describing the relationships between daily transpiration and GPP as a function of daily water vapor deficit and incoming solar radiation. It is well written paper and can be, from my point of view, very interesting for BGS readers especially in respect to applied data analysis and interpretation of the evapotranspiration - GPP relationships.

Suggested model was tested via results of eddy covariance flux measurements at 110 FLUXNET measuring sites. Most sites are located in the areas with temperate climate conditions. A limited data were used from the areas with dry (arid and semiarid) climates (according to list of sites given in supplement). It is pity that the tropical area was presented by one site only (climate of tropical savannas in North Australia). No data available from the areas with tropical rainforest or monsoon climate. It is known that there are very limited flux data for the areas. But, what is author opinion, is it possible

to apply suggested WUE model to large tropical areas and especially to tropical rain and monsoon forests? Or not?

Other point. Is there any difference in found transpiration - GPP relationships between forest and grassland sites? It is important to know taking into account the differences in plant canopy architecture of the forest and grassland sites (as well as penetration of solar radiation, stomatal regulation, etc.). Any way this point should be additionally discussed.

Specific comments.

Page 4, line 6. Sentence is not complete.

Page 4, line 16. Index uWUE is not explained.

Page 7, line 15. Index STO is not explained.

Page 9, line 14. What is about dew formation and its evaporation? Is it ignored?

Page 9, line 17. Did you analyze the relationships between contributions of soil evaporation to ET and canopy LAI?

Page 16 line 29. Term "underlying water-use efficiency (uWUE)" must be explained in page 6.

Conclusion is absent.

---

## Referee Comment (RC3) · Anonymous Referee #3 · 5 Feb 2017

The authors introduced a new term into they called "water use efficiency model" which was from Zhou et al., (2014). The rationale is that by fitting ET with GPP using this model, they find bias in many sites and the seasonality of this bias is consistent with global radiation, and global radiation can directly affect transpiration independence of VPD. This added term can explain up to half of the predicted transpiration. This manuscript reads well and is within the scope of Biogeosciences. However, I have some comments to reduce its importance.

General comments:

Introduction: Ambiguity in concepts such as WUE models, physiological WUE models? Are the authors talking about stomatal conductance models? Please clarify them and provide details. Several statements are confusing and sometimes incorrect. A few

others need reference. Please see my specific comments below.

Methods: I have concerns about the proposed models and selection criteria for the best model. The concerns include whether and how the authors test the collinearity between the variables such as Rg and GPP*VPD0.5 in the model fitting; how the authors deal with the interactive terms among those variables; In addition to MEF, index such as AIC or AICc are needed to account for possible Over-parameterization? The findings are likely to change due to different ways to construct and evaluate the models. Please also see my specific comments below.

Results: In results, in addition to the MEF, I would like to see the distribution of two other parameters (uwue and r) of all the sites. Also see my specific comments below. It would be great if the authors could provide specific data used in this study for interested readers to reproduce these results.

Discussion: All the proposed models have their own assumptions and their possible violations. Please discuss them as well on how these violations could affect the results.

Specific comments:

p1, line 19: global change instead of Global Change

p2, lines 5-15: this paragraph needs to clarify the difference between existing WUE models. According to my understanding, Beer et al., 2009 assume the ratio of ci/ca is constant and therefore derives WUE ∝ r/VPD; whereas in Zhou et al., (2014, 2015), they set up ci/ca depending on square-rooted VPD.

P2, line16: what is physiological WUE models? Did the authors mean stomatal conductance models?

P2, line 19: I may misunderstand this sentence. I thought the conductance term g0 is in the stomatal conductance models in ecosystem models (de Kauwe et al., 2013).

P2, lines 22-29: there are several confusing/incorrect statements in this paragraph.

"The models implicitly assume that,..., the ratio of both is constant with regard to this factor": the ratio GPP/ET is never constant and is considered to be proportional to vpd or squared rooted vpd depending on assumptions (Zhou et al., 2014). Line 26: I did not find the result in Leuning et al., 2008 about the nonlimiting behavior of PET to radiation. "This is a second process... cannot accommodate": again I do not think it is a true statement (see de Kauwe et al., 2013). Reference is needed to back up the statements in the first two paragraphs.

P2, line 10 Baldocchi et al., 2001 is not an appropriate reference for La Thuile database. Please cite this source: FLUXNET Synthesis Dataset (La Thuile 2007). Available at www.fluxdata.org

P4, line 6 the sentence was not complete.

P4, line 8: Not sure how water limitation can affect collinearity of parameters? Please clarify.

P4, lines 9-10: please provide reference for this statement.

P5 line 12-15: the authors tried to test the possible interactions between the two additional variables using equation (6). Would not it be an interactive term such as VPD*Rg in the equation? Please clarify.

P5, lines 24-28: More details are needed on the variance and covariance for each of the variables including GPP and ET, because this variance and covariance directly affect your L-M algorithm and likely results.

P5, line 30: an AIC or AICc index is more appropriate than the MEF, because there is likely over-parameterization of your models. Please provide these index in addition to MEF.

P7, line 15: STO? Typo?

P9, line 19-21. This is interesting finding. Could it be possible for the authors to provide

this similar figure for each of the sites in the supplementary materials for the readers to eyeball the site difference or similarity?

P16 line 19: . . . degree on variables other?? Truncated sentence?

P17 line 2: the cited paper Keenan et al., 2013 is not listed in reference list.

  Reference

Kauwe, Martin G., et al. "Forest water use and water use efficiency at elevated CO2: a model‐data intercomparison at two contrasting temperate forest FACE sites." Global Change Biology 19.6 (2013): 1759-1779.

Keenan, Trevor F., et al. "Increase in forest water-use efficiency as atmospheric carbon dioxide concentrations rise." Nature 499.7458 (2013): 324-327.

---

## Author Comment (AC1) · 31 Mar 2017

We thank Referee #1 for the positive and constructive appraisal of our article! Below, we respond to the general and specific points of the review.

METRICS FOR MODEL EVALUATION "But MEF does not provide a quantitative measure of the increase in the amount of variance explained by the successive modiïfications to the model. Table 2 could contain such an estimation, based i.e. on the (average) changes in the RMSE." We agree that the Nash-Sutcliffe Efficiency (MEF) captures only a part of the overall model performance. Specifically, the absolute magnitude of the errors is not reflected in this measure. We therefore added an complementing table for the cross-validated RMSE (Fig. 1 of this response) to the supplementary materials.

NEGATIVE INTERCEPTS "The only problem in the model definition is that the intercept is left to be negative, which has no biological meaning [...]" We also think that negative intercepts are biologically implausible. A squared intercept caused problems in the optimization, but it was possible to use a constrained optimization that limits the intercept parameter range to positive values. As can be seen in the attached plot (Fig. 2 of this response), the differences in MEF between the original model +ETres and the bounded variant +ETres_bnd is miniscule. Because it precludes biologically implausible values, we revised all relevant plots for the manuscript to account for this.

NONLINEARITY "It remains unclear why the effect of the radiation was modelled as a linear" This is a valid point! We previously tested nonlinear models, but did not detect any notable effect on model performance. However, the decision to only address linear models can appear ad hoc. We now refer to the suggested "observed–predicted" (see below) plots to support our choice of a linear response to radiation. We also included a nonlinear model variant ("+Rg_nl") of the form ET = (GPP*VPD^0.5 / uWUE) + r*RG^a, where a = 0.7 in Fig. 2. This could represent a possible, gradual saturation effect for higher radiation levels. However, including this nonlinear response yielded inferior results.

ILLUSTRATING THE IMPROVEMENT OF THE MODELS "Overall the manuscript has the tendency to not display the data and the relationships between them. Showing, for a couple of examples, the gain in having an intercept and incorporating radiation in the modelling would be great." We concur that the previous version of the manuscript did not sufficiently show the actual predictions but rather metrics reflecting the skill of the employed models. We selected two sites for which we detail how the different model formulations affect the predictions. We originally plotted this as time series. However, the difference between the models were hard to extract, as our selection of data points for the estimation yields a very sparse and irregularly sampled time series. Therefore, we propose to plot observed against predicted values with a one-to-one line indicating perfect model fit. Two such plots, with two subplots for +ETres and +Rad respectively

are attached to this reply (Fig. 3a,b of this response).

FRACTION OF SIGNIFICANT INTERCEPTS "How many times (or in percent) has the intercept ETres been found signïfi-cant?" This is an important number that is now part of the revised manuscript. The estimated intercepts were significant at 86 sites, which is 78% of all 110 sites considered. We added this number in the appropriate place in the manuscript.

In addition, all specific points referring to spelling, coherence, terminology and citations were considered and integrated in the revised manuscript.

Thank you again for your assistance in improving this paper!

———————————————————————

| model | Zhou | +ETres | +VPD | +Rg | +VPD+Rg |
|---|---|---|---|---|---|
| Zhou | NA | 0.06 | 0.08 | 0.15 | 0.16 |
| +ETres | -0.06 | NA | 0.02 | 0.09 | 0.1 |
| +VPD | -0.08 | -0.02 | NA | 0.07 | 0.08 |
| +Rg | -0.15 | -0.09 | -0.07 | NA | 0.01 |
| +VPD+Rg | -0.16 | -0.1 | -0.08 | -0.01 | NA |

**Fig. 1.** Table of mean differences of cross-validated RMSEs across all 110 sites.

[Figure]

[Figure]

**Fig. 2.** Cross-validated model-efficiencies with three additional model variants: +ETres_bnd (pos. parameters), +Rg_n (nonlinear Rg) and Interaction variant (req. by Ref #3).

[Figure]

**Fig. 3.** Predicted vs. observed ET for the FLUXNET site BE-Lon for the +ETres and +Rg model variants, respectively.

[Figure]

**Fig. 4.** Predicted vs. observed ET for the FLUXNET site HU-Bug for the +ETres and +Rg model variants, respectively.

---

## Author Comment (AC2) · 31 Mar 2017

We thank Referee #2 for the positive and constructive appraisal of our article! Below, we respond to the general and specific points of the review.

TROPICAL CLIMATES AND ECOSYSTEMS "But, what is author opinion, is it possible to apply suggested WUE model to large tropical areas and especially to tropical rain and monsoon forests? Or not?" For all its extensive coverage of northern latitudes, the FLUXNET is limited regarding a representation of tropical sites. This exacerbated by our choice of sites with a "Fair and Free Use" data policy. The only tropical site in our pool was the Australian site AU-How. However, we surmise that the observed effects are not restricted to extra-tropical regions, as the potential factors driving them are likely of physical nature and not due to specific processes limited to certain plant

types or ecosystems. Yet one potential limitation for tropical areas with frequent precipitation events is the necessary filtering for successively rain-free periods. This would, of course, be less a problem for the mentioned monsoon climates with pronounced dry-seasons. The revised manuscript now alludes to how our inferences can only be cautiously extrapolated beyond the coverage of our subset of FLUXNET sites.

EFFECT OF VEGETATION STRUCTURE "Is there any difference in found transpiration - GPP relationships between forest and grassland sites?" Referee #2 raises an important question regarding potential differences between different kinds of vegetation structures, such as grasslands and forests. Following up on this suggestion, we partitioned all sites in two classes: Low vegetation for grasslands, crops and savannas and high vegetation for all other vegetation types. When stratifying the data set like this, we found that uWUE was not significantly different for either vegetation type (Fig. 1 of this response). However, we noted that grasslands and crops had a significantly higher mean value of r (Fig. 2 of this response). This is a relevant finding, as it supports our proposed explanation that the radiation effect could be a sign of equilibrium transpiration (Jarvis and McNaughton, 1986). In a preceding study, McNaughton and Jarvis (1983) report that grasslands had a higher decoupling parameter $\Omega$, which quantifies the contribution of equilibrium evaporation. As Jarvis and McNaughton (1986) discuss, a stronger atmospheric decoupling (high $\Omega$) implies a higher relative share of equilibrium transpiration. Therefore, we repeated the analysis of the fraction of radiation-associated transpiration (Fig. 3 of this response) and found that this metric, ET_frac, was significantly higher for the low vegetation PFTs grassland and crops (0.53, 95% CI: 0.48-0.58) compared to high vegetation (0.39, 95% CI: 0.34-0.44). We revised and adapted the manuscript accordingly!

DEWFALL "What is about dew formation and its evaporation? Is it ignored?" We in fact did not consider dewfall in the original article. The evaporation of dewfall would likely be very dependent on radiation, thereby violating our assumption that the observed latent heat flux represents transpiration and hence confounding our estimates of radiation

sensitivity. To verify that our results were not biased by potentially including days with dewfall, we stratified the data set according to the relative humidity during night-time. As high relative humidity is a necessary, but not a sufficient criterion for dewfall, we consider this a conservative rule for which days are likely unaffected by any dewfall. We then estimated the term ET_res separately for all sites that had had observations for the respective RH intervals (Fig. 4 of this response), finding that the mean ETres for all sites was insensitive to this variable. We also found that excluding days with high relative humidity caused only very minor changes in the mean cross-validated model-efficiencies.

EFFECT OF LAI "Did you analyze the relationships between contributions of soil evaporation to ET and canopy LAI? LAI is an important factor for models of evapotranspiration. With the selection of rain-free periods, we assumed that both interception and bare-soil evaporation would be negligible in our analysis. To verify that our observed patterns are nevertheless merely the result of open canopies, we performed an analysis in which we filtered for successively higher LAI observations. In the corresponding figure (Fig. 5 of this response), we show the smoothed response of mean MEF over all sites for both the Zhou and the +Rg model. As the difference in performance actually *widens* for higher LAI values, we conclude that the observed patterns are unlikely the product of open vs. closed canopies. As we discuss in a preceding point, we have now better support for the explanation involving equilibrium transpiration.

In addition, all specific points referring to spelling, coherence and citations were considered and integrated in the revised manuscript.

Thank you again for your assistance in improving this paper!
* * *
[Figure]

**Fig. 1.** Parameter distribution for sites separated by vegetation structure for the parameter uWUE.

[Figure]

[Figure]

**Fig. 2.** Parameter distribution for sites separated by vegetation structure for the parameter r.

[Figure]

[Figure]

**Fig. 3.** Distribution of the fraction of ET associated with Rg separated by vegetation structure.

**Fig. 4.** Distribution of the ETres parameter stratified according to relative humidity during night-time.

**Fig. 5.** Response of the model-efficiency for the Zhou (here referred to as Kat) and the +Rg (Here referred to as Kat+Rad) model variants to increasing filtering for higher LAI

---

## Author Comment (AC3) · 31 Mar 2017

We thank Referee #3 for the positive and constructive appraisal of our article! Below, we respond to the general and specific points of the review.

AMBIGUITY OF THE MODELS. "Ambiguity in concepts such as WUE models, physiological WUE models? Are the authors talking about stomatal conductance models? Please clarify them and provide details." Thank you for pointing this out. The revised manuscript gives a more detailed introduction into the different types of WUE models and treats the terminology more carefully.

COLLINEARITY "The concerns include whether and how the authors test the collinearity between the variables such as Rg and GPP*VPD0.5 in the model fitting" This is a very good remark! The high degree of correlation is an important issue for these kind

of empirical analyses. This is particularly pertinent for isolating the fraction of evapotranspiration that we attribute to radiation. In the original paper, we analyzed the impact of collinearity on our results by accounting for the correlation of parameter uncertainties (Supplementary Materials S1). In the new manuscript the problem of collinearity and our treatment is given more prominence. We further moved the mentioned section from the supplement to the method section of the main document.

OVERPARAMETERIZATION "In addition to MEF, index such as AIC or AICc are needed to account for possible over-parameterization?" We fully agree with Referee #3 that overparameterization is an important issue in analyses focussed on model selection. We believe that we have adequately addressed this problem by exclusively using cross-validated Nash-Sutcliffe Efficiencies (MEF) in our model comparison. Adding further parameters to a model will generally allow the model to accommodate even observations that were the result of random errors or unattributed processes. The cross-validation penalizes such a over-parameterization by iteratively testing the model's ability to predict observations that it wasn't calibrated to. Using an information criterion, such as AIC, AICc or BIC, that directly accounts for the number of parameters used in the model is another possibility to represent model complexity. AICc can be expected to converge with cross-validation asymptotically (Stone, 1977). We are therefore confident that our results are not confounded by the number of model parameters. To illustrate this, we added a table to this comment (Fig. 1 of this response) that replicates Table 1 of the original paper (Fraction of sites with a higher or lower MEF). As is the appropriate usage for AICc, we counted the fraction of sites were the pairwise difference of AICc was smaller than -2 for the model of the row to be considered superior to that of the corresponding column. The table suggests that our conclusions are not sensitive to the choice of either AICc or cross-validated MEFs.

INTERACTION TERMS "How the authors deal with the interactive terms among those variables." This is an interesting question. In our analysis we aimed to obtain effective, parsimonious models with sufficient biological and physical plausibility. This is why we

did not test all possible combinations of predictor variables. Attached to this comment is a plot (Fig. 2 of this response) that includes both the three models of the old manuscript (Zhou, +ETres, +Rg) and three new variants for questions raised by the other referees: +ETres_bnd has parameters constrained to positive values, +Rg_nl has a nonlinear response to radiation and Intrct has an interaction term of VPD with Rg. The model evaluation was performed in a comprehensive cross-validation scheme. The original +Rg variant was again confirmed to have the highest performance in this evaluation. In addition, this model is corroborated by the new results indicating higher importance of radiation for low vegetation, which makes equilibrium evaporation a plausible candidate explanation for the observed patterns (see below in our response, section PARAME-TER DISTRIBUTIONS).

INTRODUCTORY DEFINITION OF MODELS - "p2, lines 5-15: this paragraph needs to clarify the difference between existing WUE models." - "P2, lines 22-29: there are several confusing/incorrect statements in this paragraph." - "P2, line16: what is phys-iological WUE models? Did the authors mean stomatal conductance models?" Thank you for pointing this out! We have revised the introduction of the paper accordingly, to better explain our approach and contrast it with existing models. We also discus how current models include the g0 conductance term. "the ratio GPP/ET is never constant and is considered to be proportional to vpd or squared rooted vpd depending on as-sumptions (Zhou et al., 2014)" Here, we referred to radiation, when stating that "The models implicitly assume that, at ecosystem-scale, GPP and ET respond equally to changes in radiation and that, therefore, the ratio of both is constant with regard to this factor." This has been clarified in the revised introduction.

EFFECT OF WATER-LIMITATION ON COLLINEARITY "Not sure how water limitation can affect collinearity of parameters?" Referee #3 is right that we did not provide a sufficient explanation for our reasoning here. As we mention below, the degree of cor-relation between the predictor variables is a property of the additive models we iden-tified. The dependency of GPP on radiation is an obvious case for that. We expected

this correlation (and the following collinearity) to decrease under water-limitation, as GPP is then no longer as easily determined by radiation. For example, during periods of extended droughts, we would expect day-to-day variability of GPP to be no longer a function of radiation and related covariates but rather variables reflecting soil-water availability. If this dependency of the covariates decreases, it would follow that the collinearity of the parameters decreases, too. However, the results were very inconclusive when adopting the aridity index (AI) that is used by the United Nations Environmental Program (defined as: AI = Precipitation / PET). We decided that the results were furthermore not pertinent to the main topic, which is why we decided to exclude this part from the new manuscript.

PARAMETER DISTRIBUTIONS "In results, in addition to the MEF, I would like to see the distribution of two other parameters (uwue and r) of all the sites." The updated manuscript now includes plots showing the distribution of the two parameters uWUE and r. Upon a comment by Referee #2, we stratified the data-set along the vegetation structure (low for grasslands and crops, high for all other plant functional types). Quoting from our reply to Referee #1: "When stratifying the data set like this, we found that uWUE was not significantly different for either vegetation type (Kolmogorov–Smirnov test) [Fig. 3 of this response]. However, we noted that grasslands and crops had a significantly higher mean value of r [Fig. 4 of this response]. This is a relevant finding, as it supports our proposed explanation that the radiation effect could be a sign of equilibrium transpiration (Jarvis and McNaughton, 1986). In a preceding study, McNaughton and Jarvis (1983) report that grasslands had a higher decoupling parameter $\Omega$, quantifiying the contribution of equilibrium evaporation. As Jarvis and McNaughton (1986) discuss, a stronger atmospheric decoupling (high $\Omega$) implies a higher relative share of equilibrium transpiration. Therefore, we repeated the analysis of the fraction of radiation-associated transpiration and found that this metric, ET_frac, was significantly higher for the low vegetation PFTs grassland and crops (0.53, 95% CI: 0.48-0.58) compared to high vegetation (0.39, 95% CI: 0.34-0.44) [Fig. 5 of this response]. We revised and adapted the manuscript accordingly!" The relevant plots have been

attached to and renumbered for this comment (Fig. 3-5 of this response).

MONTHLY PATTERNS BY SITE "This is interesting finding. Could it be possible for the authors to provide this similar figure for each of the sites in the supplementary materials for the readers to eyeball the site difference or similarity?" The updated supplement now contains this figure as a matrix of monthly patterns for each site individually! The plot is also attached to this comment (Fig. 6).

COVARIANCE ASSUMPTIONS "More details are needed on the variance and covariance for each of the variables including GPP and ET, because this variance and covariance directly affect your L-M algorithm and likely results." The optimization approach of our analyses follows eq. 5–6 in Omlin and Reichert (1999) with a $\sigma$_meas of 1, hence being insensitive to the uncertainties in the forcing and target variables. In agreement with Lasslop et al. (2008), this approach does not consider correlations between the errors of the original latent heat and net ecosystem exchange fluxes.

LIMITATION OF THE MODELS "All the proposed models have their own assumptions and their possible violations. Please discuss them as well on how these violations could affect the results." This is a critical aspect for a model-selection exercise such as ours and is treated more diligently in the revised version of the manuscript.

DATA AVAILABILITY The data sets can be downloaded at http://fluxnet.fluxdata.org//data/download-data/

In addition, all specific points referring to spelling, coherence and citations were considered and integrated in the revised manuscript.

Thank you again for your assistance in improving this paper!

REFERENCES

Jarvis, P. G. and McNaughton, K.: Stomatal control of transpiration: scaling up from leaf to region, Advances in ecological research, 15, 49, 1986.

Lasslop, G., Reichstein, M., Kattge, J., Papale, D.: Influences of observation errors in eddy flux data on inverse model parameter estimation. Biogeosciences, 5, 1311–1324, 2008.

McNaughton, K. G., and Jarvis, P. G.. Predicting effects of vegetation changes on transpiration and evaporation. In: "Water Deficits and Plant Growth" (T. T. Kozlowski, ed.), Vol. 7, pp. 1-47. Academic Press, New York, 1983.

Omlin, M. and Reichert, P.: A comparison of techniques for the estimation of model prediction uncertainty, Ecological Modelling, 115, 45–59, 1999.

Stone, M.: An asymptotic equivalence of choice of model by cross-validation and Akaike's criterion. Journal of the Royal Statistical Society: Series B (Methodological), 39, 44–47, 1977.

––––––––––––––––––––––––––––––––––––

| model | Zhou | +ETres | +VPD | +Rg | +VPD+Rg |
|---|---|---|---|---|---|
| Zhou | NA | 0.04 | 0.03 | 0.02 | 0 |
| +ETres | 0.83 | NA | 0.34 | 0.05 | 0 |
| +VPD | 0.78 | 0.58 | NA | 0.08 | 0 |
| +Rg | 0.96 | 0.93 | 0.89 | NA | 0.05 |
| +VPD+Rg | 0.98 | 0.99 | 0.94 | 0.58 | NA |

**Fig. 1.** Fraction of sites that had an delta-AICc smaller than -2 when comparing a model of a given row with a given column

[Figure]

**Fig. 2.** Distribution of cross-validated model-efficiencies with three additional models (positive bounded parameters, nonlinear Rg response and a VPD-Rg interaction effect).

[Figure]

**Fig. 3.** Distribution of the parameter uWUE for all sites separated by vegetation structure.

**Fig. 4.** Distribution of the parameter r for all sites separated by vegetation structure.

[Figure]

**Fig. 5.** Distribution of the fraction of radiation-associated ET for all sites separated by vegetation structure.

[Figure]

[Figure]

**Fig. 6.** Month-wise estimates of ETres for all sites included in the analysis.

---

## Author Comment (AC4) · 31 Mar 2017

REFERENCES

Jarvis, P. G. and McNaughton, K.: Stomatal control of transpiration: scaling up from leaf to region, Advances in ecological research, 15, 49, 1986.

McNaughton, K. G., and Jarvis, P. G.. Predicting effects of vegetation changes on transpiration and evaporation. In: "Water Deficits and Plant Growth" (T. T. Kozlowski, ed.), Vol. 7, pp. 1-47. Academic Press, New York, 1983.

---

## Author Response (AR1)

**AUTHOR RESPONSE**

The manuscript for our paper *The importance of radiation for semi-empirical water-use efficiency models* has been revised according to the reviews. We carefully considered the suggestions made by three anonymous referees. We integrate both the minor comments regarding spelling, coherence and citations, as well as general comments made by the three referees. The proposed changes have improved the manuscript substantially, both in its intelligibility and its implications.

The major change has been the addition of a stratification of all included sites by type of vegetation, as suggested by Referee #2. The revised manuscript now addresses differences between low and high vegetation types for the two parameters uWUE and *r*, as well as the metric ET_frac. The results of these analyses have given additional credence to the proposed explanation that equilibrium evaporation is responsible for observed effects of radiation on transpiration.

Substantial changes have been made to the supplementary materials, which now includes multiple suggestions made by the referees.

The introduction has been revised for improved clarity.

Upon consideration, we think that model limitations are adequately addressed in our discussion section. To highlight this important part, we introduced a new subsection, "4.2 Limitations".

Below, you can find our responses to the three referees, as we have made published them for each referee comment (see the respective DOI for the full comments that include the figures).

Thank you for your assistance in improving this manuscript.

Best,
Sven Boese on behalf of the co-authors

**Referee 1**

```
Please refer to our original response to Referee #1 for the full
author comment including the referenced figures.
AC1:   doi:10.5194/bg-2016-524-AC1
```

We thank Referee #1 for the positive and constructive appraisal of our article! Below, we respond to the general and specific points of the review.

**Metrics For Model Evaluation**

"But MEF does not provide a quantitative measure of the increase in the amount of variance explained by the successive modifications to the model. Table 2 could contain such an estimation, based i.e. on the (average) changes in the RMSE." We agree that the Nash-Sutcliffe Efficiency (MEF) captures only a part of the overall model performance. Specifically, the absolute magnitude of the errors is not reflected in this measure. We therefore added an complementing table for the cross-validated RMSE (Fig. 1 of this response) to the supplementary materials.

**Negative Intercepts**

"The only problem in the model definition is that the intercept is left to be negative, which has no biological meaning [...]" We also think that negative intercepts are biologically implausible. A squared intercept caused problems in the optimization, but it was possible to use a constrained optimization that limits the intercept parameter range to positive values. As can be seen in the attached plot (Fig. 2 of this response), the differences in MEF between the original model +ETres and the bounded variant +ETres_bnd is miniscule. Because it precludes biologically implausible values, we revised all relevant plots for the manuscript to account for this.

**Nonlinearity**

"It remains unclear why the effect of the radiation was modelled as a linear" This is a valid point! We previously tested nonlinear models, but did not detect any notable effect on model performance. However, the decision to only address linear models can appear ad hoc. We now refer to the suggested "observed–predicted" (see below) plots to support our choice of a linear response to radiation. We also included a nonlinear model variant ("+Rg_nl") of the form ET =

(GPP$VPD^{0.5}$ / uWUE) + rRG$^{a}$, where a = 0.7 in Fig. 2. This could represent a possible, gradual saturation effect for higher radiation levels. However, including this nonlinear response yielded inferior results.

**Illustrating the Improvement of the Models**

"Overall the manuscript has the tendency to not display the data and the relationships between them. Showing, for a couple of examples, the gain in having an intercept and incorporating radiation in the modelling would be great." We concur that the previous version of the manuscript did not sufficiently show the actual predictions but rather metrics reflecting the skill of the employed models. We selected two sites for which we detail how the different model formulations affect the predictions. We originally plotted this as time series. However, the difference between the models were hard to extract, as our selection of data points for the estimation yields a very sparse and irregularly sampled time series. Therefore, we propose to plot observed against predicted values with a one-to-one line indicating perfect model fit. Two such plots, with two subplots for +ETres and +Rad respectively are attached to this reply (Fig. 3a,b of this response).

**Fraction of Significant Intercepts**

"How many times (or in percent) has the intercept ETres been found signifi-cant?" This is an important number that is now part of the revised manuscript. The estimated intercepts were significant at 86 sites, which is 78% of all 110 sites considered. We added this number in the appropriate place in the manuscript.

In addition, all specific points referring to spelling, coherence, terminology and citations were considered and integrated in the revised manuscript.

Thank you again for your assistance in improving this paper!

**Referee 2**

```
Please refer to our original response to Referee #2 for the full
author comment including the referenced figures.
AC2: doi:10.5194/bg-2016-524-AC2
AC4: doi:10.5194/bg-2016-524-AC4 (missing references)
```

We thank Referee #2 for the positive and constructive appraisal of our article!
Below, we respond to the general and specific points of the review.

**Tropical Climates and Ecosystems**

"But, what is author opinion, is it possible to apply suggested WUE model to large
tropical areas and especially to tropical rain and monsoon forests? Or not?" For all
its extensive coverage of northern latitudes, the FLUXNET is limited regarding a
representation of tropical sites. This exarcerbated by our choice of sites with a
"Fair and Free Use" data policy. The only tropical site in our pool was the
Australian site AU-How. However, we surmise that the observed effects are not
restricted to extra-tropical regions, as the potential factors driving them are likely
of physical nature and not due to specific processes limited to certain plant types
or ecosystems. Yet one potential limitation for tropical areas with frequent
precipitation events is the necessary filtering for successively rain-free periods.
This would, of course, be less a problem for the mentioned monsoon climates
with pronounced dry-seasons. The revised manuscript now alludes to how our
inferences can only be cautiously extrapolated beyond the coverage of our subset
of FLUXNET sites.

**Effect of Vegetation Structure**

"Is there any difference in found transpiration - GPP relationships between forest
and grassland sites?" Referee #2 raises an important question regarding potential
differences between different kinds of vegetation structures, such as grasslands
and forests. Following up on this suggestion, we partitioned all sites in two
classes: Low vegetation for grasslands, crops and savannas and high vegetation
for all other vegetation types. When stratifying the data set like this, we found
that uWUE was not significantly different for either vegetation type (Fig. 1).
However, we noted that grasslands and crops had a significantly higher mean
value of $r$ (Fig. 2). This is a relevant finding, as it supports our proposed
explanation that the radiation effect could be a sign of equilibrium transpiration
(Jarvis and McNaughton, 1986). In a preceding study, McNaughton and Jarvis
(1983) report that grasslands had a higher decoupling parameter $\Omega$, which
quantifies the contribution of equilibrium evaporation. As Jarvis and McNaughton
(1986) discuss, a stronger atmospheric decoupling (high $\Omega$) implies a higher
relative share of equilibrium transpiration. Therefore, we repeated the analysis of
the fraction of radiation-associated transpiration (Fig. 3) and found that this
metric, ET_frac, was significantly higher for the low vegetation PFTs grassland
and crops (0.53, 95% CI: 0.48–0.58) compared to high vegetation (0.39, 95% CI:
0.34–0.44). We revised and adapted the manuscript accordingly!

**Dewfall**

"What is about dew formation and its evaporation? Is it ignored?" We in fact did not consider dewfall in the original article. The evaporation of dewfall would likely be very dependent on radiation, thereby violating our assumption that the observed latent heat flux represents transpiration and hence confounding our estimates of radiation sensitivity. To verify that our results were not biased by potentially including days with dewfall, we stratified the data set according to the relative humidity during night-time. As high relative humidity is a necessary, but not a sufficient criterion for dewfall, we consider this a conservative rule for which days are likely unaffected by any dewfall. We then estimated the term ET_res separately for all sites that had had observations for the respective RH intervals (Fig. 4), finding that the mean ETres for all sites was insensitive to this variable. We also found that excluding days with high relative humidity caused only very minor changes in the mean cross-validated model-efficiencies.

**Effect of LAI**

"Did you analyze the relationships between contributions of soil evaporation to ET and canopy LAI? LAI is an important factor for models of evapotranspiration. With the selection of rain-free periods, we assumed that both interception and bare-soil evaporation would be negligible in our analysis. To verify that our observed patterns are nevertheless merely the result of open canopies, we performed an analysis in which we filtered for successively higher LAI observations. In the corresponding figure (Fig. 5), we show the smoothed response of mean MEF over all sites for both the Zhou and the +Rg model. As the difference in performance actually *widens* for higher LAI values, we conclude that the observed patterns are unlikely the product of open vs. closed canopies. As we discuss in a preceding point, we have now better support for the explanation involving equilibrium transpiration.

In addition, all specific points referring to spelling, coherence and citations were considered and integrated in the revised manuscript.

Thank you again for your assistance in improving this paper!

```

We thank Referee #3 for the positive and constructive appraisal of our article! Below, we respond to the general and specific points of the review.

**Ambiguity of the Models.**

"Ambiguity in concepts such as WUE models, physiological WUE models? Are the authors talking about stomatal conductance models? Please clarify them and provide details." Thank you for pointing this out. The revised manuscript gives a more detailed introduction into the different types of WUE models and treats the terminology more carefully.

**Collinearity**

"The concerns include whether and how the authors test the collinearity between the variables such as Rg and GPP*VPD0.5 in the model fitting" This is a very good remark! The high degree of correlation is an important issue for these kind of empirical analyses. This is particularly pertinent for isolating the fraction of evapotranspiration that we attribute to radiation. In the original paper, we analyzed the impact of collinearity on our results by accounting for the correlation of parameter uncertainties (Supplementary Materials S1). In the new manuscript the problem of collinearity and our treatment is given more prominence. We further moved the mentioned section from the supplement to the method section of the main document.

**Overparameterization**

"In addition to MEF, index such as AIC or AICc are needed to account for possible over-parameterization?" We fully agree with Referee #3 that overparameterization is an important issue in analyses focussed on model selection. We believe that we have adequately addressed this problem by

exclusively using cross-validated Nash-Sutcliffe Efficiencies (MEF) in our model comparison. Adding further parameters to a model will generally allow the model to accommodate even observations that were the result of random errors or unattributed processes. The cross-validation penalizes such a over-parameterization by iteratively testing the model's ability to predict observations that it wasn't calibrated to. Using an information criterion, such as AIC, AICc or BIC, that directly accounts for the number of parameters used in the model is another possibility to represent model complexity. AICc can be expected to converge with cross-validation asymptotically (Stone, 1977). We are therefore confident that our results are not confounded by the number of model parameters. To illustrate this, we added a table to this comment (Fig. 1 of this response) that replicates Table 1 of the original paper (Fraction of sites with a higher or lower MEF). As is the appropriate usage for AICc, we counted the fraction of sites were the pairwise difference of AICc was smaller than −2 for the model of the row to be considered superior to that of the corresponding column. The table suggests that our conclusions are not sensitive to the choice of either AICc or cross-validated MEFs.

**Interaction Terms**

"How the authors deal with the interactive terms among those variables." This is an interesting question. In our analysis we aimed to obtain effective, parsimonious models with sufficient biological and physical plausibility. This is why we did not test all possible combinations of predictor variables. Attached to this comment is a plot (Fig. 2 of this response) that includes both the three models of the old manuscript (Zhou, +ETres, +Rg) and three new variants for questions raised by the other referees: +ETres_bnd has parameters constrained to positive values, +Rg_nl has a nonlinear response to radiation and Intrct has an interaction term of VPD with Rg. The model evaluation was performed in a comprehensive cross-validation scheme. The original +Rg variant was again confirmed to have the highest performance in this evaluation. In addition, this model is corroborated by the new results indicating higher importance of radiation for low vegetation, which makes equilibrium evaporation a plausible candidate explanation for the observed patterns (see below in our response, section PARAMETER DISTRIBUTIONS).

**Introductory Definition of Models**
– "p2, lines 5–15: this paragraph needs to clarify the difference between existing WUE models."
– "P2, lines 22–29: there are several confusing/incorrect statements in this

paragraph."

– "P2, line16: what is physiological WUE models? Did the authors mean stomatal conductance models?" Thank you for pointing this out! We have revised the introduction of the paper accordingly, to better explain our approach and contrast it with existing models. We also discus how current models include the g0 conductance term. "the ratio GPP/ET is never constant and is considered to be proportional to vpd or squared rooted vpd depending on assumptions (Zhou et al., 2014)" Here, we referred to radiation, when stating that "The models implicitly assume that, at ecosystem-scale, GPP and ET respond equally to changes in radiation and that, therefore, the ratio of both is constant with regard to this factor." This has been clarified in the revised introduction.

**Effect of Water-Limitation on Collinearity**

"Not sure how water limitation can affect collinearity of parameters?" Referee #3 is right that we did not provide a sufficient explanation for our reasoning here. As we mention below, the degree of correlation between the predictor variables is a property of the additive models we identified. The dependency of GPP on radiation is an obvious case for that. We expected this correlation (and the following collinearity) to decrease under water-limitation, as GPP is then no longer as easily determined by radiation. For example, during periods of extended droughts, we would expect day-to-day variability of GPP to be no longer a function of radiation and related covariates but rather variables reflecting soil-water availability. If this dependency of the covariates decreases, it would follow that the collinearity of the parameters decreases, too. However, the results were very inconclusive when adopting the aridity index (AI) that is used by the United Nations Environmental Program (defined as: AI = Precipitation / PET). We decided that the results were furthermore not pertinent to the main topic, which is why we decided to exclude this part from the new manuscript.

**Parameter Distributions**

"In results, in addition to the MEF, I would like to see the distribution of two other parameters (uwue and r) of all the sites." The updated manuscript now includes plots showing the distribution of the two parameters uWUE and r. Upon a comment by Referee #2, we stratified the data-set along the vegetation structure (low for grasslands and crops, high for all other plant functional types). Quoting from our reply to Referee #1: "When stratifying the data set like this, we found that uWUE was not significantly different for either vegetation type (Kolmogorov–Smirnov test) [Fig. 3 of this response]. However, we noted that grasslands and crops had a significantly higher mean value of r [Fig. 4 of this response]. This is a

relevant finding, as it supports our proposed explanation that the radiation effect could be a sign of equilibrium transpiration (Jarvis and McNaughton, 1986). In a preceding study, McNaughton and Jarvis (1983) report that grasslands had a higher decoupling parameter Ω, quantifiying the contribution of equilibrium evaporation. As Jarvis and McNaughton (1986) discuss, a stronger atmospheric decoupling (high Ω) implies a higher relative share of equilibrium transpiration. Therefore, we repeated the analysis of the fraction of radiation-associated transpiration and found that this metric, ET_frac, was significantly higher for the low vegetation PFTs grassland and crops (0.53, 95% CI: 0.48–0.58) compared to high vegetation (0.39, 95% CI: 0.34–0.44) [Fig. 5 of this response]. We revised and adapted the manuscript accordingly!" The relevant plots have been attached to and renumbered for this comment (Fig. 3–5 of this response).

**Monthly Patterns by Site**

"This is interesting finding. Could it be possible for the authors to provide this similar figure for each of the sites in the supplementary materials for the readers to eyeball the site difference or similarity?" The updated supplement now contains this figure as a matrix of monthly patterns for each site individually! The plot is also attached to this comment (Fig. 6).

**Covariance Assumptions**

"More details are needed on the variance and covariance for each of the variables including GPP and ET, because this variance and covariance directly affect your L-M algorithm and likely results." The optimization approach of our analyses follows eq. 5–6 in Omlin and Reichert (1999) with a σ_meas of 1, hence being insensitive to the uncertainties in the forcing and target variables. In agreement with Lasslop et al. (2008), this approach does not consider correlations between the errors of the original latent heat and net ecosystem exchange fluxes.

**Limitation of the Models**

"All the proposed models have their own assumptions and their possible violations. Please discuss them as well on how these violations could affect the results." This is a critical aspect for a model-selection exercise such as ours and is treated more diligently in the revised version of the manuscript.

**Data Availability**

The data sets can be downloaded at http://fluxnet.fluxdata.org//data/download-data/

In addition, all specific points referring to spelling, coherence and citations were considered and integrated in the revised manuscript.

Thank you again for your assistance in improving this paper!

[revised manuscript text omitted]

---

## Author Response (AR2)

Dear Dr. Ito,

Thank you for the positive evaluation of our paper "The importance of radiation for semi-empirical water-use efficiency models". The points that were mentioned in the final editor report have been corrected.

We noted a minor mistake which we corrected in the final version of the article. The Table S1 listing all FLUXNET sites that we used included two sites that were not used in our analysis. The table listed 112 sites, in contrast to the correct number of 110 as stated throughout the paper. We corrected this mistake in the updated version of the supplement.

We made minor changes to the figures, such as consistent use of subscripts for the unit *grams of carbon* and added spaces between units were necessary.

Thank you again for your assistance in improving this document.

Best regards

Sven Boese on behalf of the coauthors